# The Potential for Renal Injury Elicited by Physical Work in the Heat

**DOI:** 10.3390/nu11092087

**Published:** 2019-09-04

**Authors:** Zachary J. Schlader, David Hostler, Mark D. Parker, Riana R. Pryor, James W. Lohr, Blair D. Johnson, Christopher L. Chapman

**Affiliations:** 1Center for Research and Education in Special Environments, Department of Exercise and Nutrition Sciences, University at Buffalo, Buffalo, NY 14214, USA; 2Department of Kinesiology, School of Public Health, Indiana University, Bloomington, IN 47405, USA; 3Department of Physiology and Biophysics, Jacobs School of Medicine and Biomedical Sciences, University at Buffalo, Buffalo, NY 14214, USA; 4Department of Ophthalmology, Jacobs School of Medicine and Biomedical Sciences, University at Buffalo, Buffalo, NY 14214, USA; 5Department of Medicine, Jacobs School of Medicine and Biomedical Sciences, University at Buffalo, Buffalo, NY 14214, USA

**Keywords:** acute kidney injury, chronic kidney disease, heat stress, dehydration, exercise

## Abstract

An epidemic of chronic kidney disease (CKD) is occurring in laborers who undertake physical work in hot conditions. Rodent data indicate that heat exposure causes kidney injury, and when this injury is regularly repeated it can elicit CKD. Studies in humans demonstrate that a single bout of exercise in the heat increases biomarkers of acute kidney injury (AKI). Elevations in AKI biomarkers in this context likely reflect an increased susceptibility of the kidneys to AKI. Data largely derived from animal models indicate that the mechanism(s) by which exercise in the heat may increase the risk of AKI is multifactorial. For instance, heat-related reductions in renal blood flow may provoke heterogenous intrarenal blood flow. This can promote localized ischemia, hypoxemia and ATP depletion in renal tubular cells, which could be exacerbated by increased sodium reabsorption. Heightened fructokinase pathway activity likely exacerbates ATP depletion occurring secondary to intrarenal fructose production and hyperuricemia. Collectively, these responses can promote inflammation and oxidative stress, thereby increasing the risk of AKI. Equivalent mechanistic evidence in humans is lacking. Such an understanding could inform the development of countermeasures to safeguard the renal health of laborers who regularly engage in physical work in hot environments.

## 1. Background

An epidemic of chronic kidney disease of unknown etiology (CKDu) is occurring in laborers who undertake physical work outdoors in hot conditions [1,2]. This disease was first described in 2002, when nephropathy was identified in a disproportionate number of young Central American sugarcane workers [3]. Thereafter, people in other occupations [4,5,6] and in other regions of the world [1,7,8] have been diagnosed with what appears to be the same disease, although there is some heterogeneity in the clinical signs, symptoms, and likely etiology [1,2]. The effects of CKDu are devastating. For instance, the Pan American Health Organization estimates that CKDu caused more than 60,000 deaths in Central America from 1997 to 2013, with 41% of these deaths occurring in people younger than 60 y of age [9]. Furthermore, the World Health Organization estimates that ~15% of workers in endemic areas are at risk of developing CKDu [10]. This estimate was recently corroborated in a meta-analysis by Flouris et al. [11]. Thus, there is an urgent need to identify effective strategies to mitigate the risk of CKDu.

The development of countermeasures for CKDu requires an understanding of the etiology of the disease. Unfortunately, the etiology underlying the development of CKDu is largely unknown [12]. Patients diagnosed with CKDu usually present with an asymptomatic rise in serum creatinine and low grade proteinuria [4,5]. Renal biopsies show interstitial fibrosis, low grade inflammation, tubular atrophy, and glomerulosclerosis with signs of ischemia [13]. The two hallmarks of CKDu are that it is not associated with traditional risk factors of chronic kidney disease (CKD) (e.g., advanced age, diabetes, hypertension) and that it is more common in laborers regularly exposed to hot environments [7,14,15,16]. There may also be a contributing role for toxin exposures (e.g., agrochemicals, heavy metals or infectious agents) [14,17] and/or the use of agents with nephrotoxic side effects (e.g., nonsteroidal anti-inflammatory drugs (NSAIDs)) [18,19]. However, convincing evidence supporting these factors as being primary in the etiology of CKDu is lacking [2,20,21]. Thus, it has been suggested that heat may be the key occupational exposure contributing to the development of CKDu [7,14,22]. This is supported by data indicating a higher prevalence of CKDu in agriculture workers exposed to hotter climates compared to those working in cooler climates, despite otherwise enduring the same occupational conditions [4,6]. As a result, it has been proposed that CKDu is a form of ‘heat stress nephropathy’, the risk of which could worsen with climate change [7,16,23]. That said, it is important to note that while factors associated with engaging in physical work in the heat are likely to play an important role, the etiology of CKDu is mostly unknown and may be multifaceted [7,24,25].

One leading etiological hypothesis for CKDu is that physical work (i.e., exercise) in the heat, which leads to heat strain (i.e., increased core body temperature) and dehydration (i.e., a hypovolemic–hyperosmotic state caused by the loss of body water due to sweating combined with inadequate fluid intake), causes acute kidney injury (AKI) [2,7,15,16,26,27,28]. This heat-related AKI is probably transient in nature [29,30,31,32], which is clinically defined as lasting <3 days [33]. Nevertheless, it has been proposed that multiple transient AKI exposures can manifest as nephropathy, affecting the renal tubules and glomeruli [7,15,16,27].

Subclinical rhabdomyolysis (i.e., muscle injury), which may occur during (or after) unaccustomed intense or prolonged exercise [34], is often proposed to contribute to heat-related AKI and CKDu [7,14,35]. This is supported by human data demonstrating that the presence of muscle damage increases the risk of AKI during exercise in the heat [36]. These data were recently corroborated in a rodent model of repeated heat exposure, which demonstrated that experimental rhabdomyolysis worsened heat-induced kidney injury [37]. That said, epidemiological data obtained from Guatemalan sugarcane workers indicate that the presence of rhabdomyolysis is not associated with cross-shift declines in kidney function [26], suggesting that the contribution of rhabdomyolysis to progressive reductions in kidney function in this population is likely small. This can be explained by the adaptation of skeletal muscle to become more resistant to damage after the initial injurious exercise [38]. Thus, while the first few days of unaccustomed intense physical work in the heat may induce subclinical rhabdomyolysis, the subsequent muscle damage incurred by manual laborers on a daily basis is probably small. Therefore, the contribution of subclinical rhabdomyolysis to heat-related AKI and CKDu is likely minimal.

It is important to note that there is currently no direct support for recurrent heat-related AKI in the etiology of CKDu [39]. For instance, to our knowledge a dose–response relation between the frequency and severity of heat-related AKI and the subsequent development of CKD has never been experimentally examined in rodent models nor explored in epidemiological studies. That said, in the absence of heat exposure, data from pre-clinical models indicate that AKI can result in renal tubular remodeling [40,41], which can lead to long-term impairments in kidney function, the defining characteristic of CKD [40,42,43,44,45,46]. More recently, these findings have been extended to instances of combined heat exposure and dehydration. For instance, rodents develop nephropathy with intermittent, repeated passive (i.e., resting) heat exposure without access to fluids over 4–5 weeks [47,48,49]. Importantly, these animals demonstrate evidence of kidney injury, which is consistent with the recurrent heat-related AKI hypothesis [47,48,49]. These data are corroborated by workplace data demonstrating increases in biomarkers of AKI both across a work shift and across the harvest season [31,32,50,51,52,53,54,55]. Notably, the working conditions during these periods are conducive to increased heat strain, dehydration, and reductions in kidney function [31,32,50,51,52,53,54,55]. Whether the observed increases in AKI biomarkers translate to an increased risk of CKD is unclear [56]. That said, epidemiological evidence clearly indicates that the frequency and severity of non-heat-related AKI is associated with the incidence and severity of CKD [57,58,59,60,61]. For instance, a single episode of relatively mild (Stage 1) AKI results in a 43% increased risk of developing advanced stage CKD within one year [57]. Furthermore, even a single episode of transient AKI that lasted ≤2 days is associated with a ~2-fold increased risk of death [62] and a 1.4-fold increased risk of developing CKD [57]. Thus, it is generally accepted that an increased frequency, severity, and/or duration of AKI elevates the risk of developing CKD [63,64], although this remains a topic of debate that requires additional exploration [44,63,64,65]. Against this background, the etiology of CKDu may be better understood by investigating the pathophysiology of the increased risk of AKI in humans exercising in the heat. This latter point is particularly important. For instance, experimentally manipulating CKDu risk in humans is unethical. However, quantification of the risk of AKI in humans may be readily accomplished when studies are carefully designed to ensure the risk of AKI is short lasting and completely resolved between experimental periods.

Despite the proposed relation between heat exposure and CKDu, the effects of heat strain and dehydration elicited by exercise in the heat on kidney health and the risk of AKI in humans is largely unexplored. We believe that this is likely because changes in kidney function induced by heat strain, exercise, and/or dehydration are believed to be physiological in nature, clinically benign, and completely reversed with recovery [17]. Emerging evidence, however, calls this dogma into question [2,14,16,36,47,51,54,66,67,68,69,70]. Therefore, the purpose of this narrative review is to present evidence that exercise in the heat may increase the risk of developing AKI in humans. In doing so, we will also address how the risk of AKI can be examined in humans and we will discuss some of the potential mechanisms underlying this risk. A focus will be placed on human subjects research, but data from non-human animals will be included where necessary. Several important knowledge gaps will also be presented. Filling these knowledge voids is vital to identifying strategies to mitigate the risk of AKI and CKDu in workers who regularly engage in physical work in hot environments.

## 2. Acute Kidney Injury

AKI refers to a clinical condition characterized by a rapid (i.e., occurring within ≤7 days) reduction in kidney function [71]. AKI can be further categorized as transient (i.e., lasting <3 days) or sustained (i.e., lasting ≥ 3 days) [33]. AKI is often reversible, but can be fatal both in the acute setting and in relation to an increased risk of CKD, which is defined as a gradual loss of kidney function that persists for >90 days [72]. Clinically, the presence and the severity of AKI is diagnosed via criteria established by international working groups such as the Acute Kidney Injury Network (AKIN) [73], the Acute Dialysis Quality Initiative (ADQI) [74], and Kidney Disease: Improving Global Outcomes (KDIGO) [71,75], amongst others. The criteria for identifying and categorizing the severity AKI differs slightly between these working groups. However, the common denominator for classifying AKI among these working groups is an acute reduction in kidney function. Changes in kidney function are often quantified via changes in glomerular filtration rate (GFR), which is widely accepted as the best overall index of kidney function in both health and disease [71], and the rate of urine production (aka: urine flow rate). Precise measurements of GFR can be cumbersome, impractical and/or costly. Thus, GFR is often estimated from the clearance of endogenous creatinine from the circulation [76]. Creatinine is formed from muscle creatine and released into the blood at a relatively constant rate provided there are no changes in muscle mass or muscle damage during the period of observation. Importantly, creatinine is not reabsorbed along the nephron tubule lumen. Correcting urinary creatinine excretion rate for serum creatinine (a systemic variable that could also influence urinary creatinine excretion) provides an accurate measure of GFR [76]. Thus, creatinine clearance is a function of urine flow rate and serum and urinary creatinine concentrations at any given time and can be calculated as:
$$GFR\approx Clearance_{\;Creatinine}=\frac{{\left[Creatinine\right]}_{urine}\times Urine\;flow\;rate}{{\left[Creatinine\right]}_{serum}}$$
where: *GFR* is glomerular filtration rate, *Clearance_Creatinine_* is creatinine clearance, [*Creatinine*]_*urine*_ is the urinary concentration of creatinine, *Urine flow rate* is the volume of urine produced per unit time, and [*Creatinine*]_*serum*_ is the serum concentration of creatinine.

Precise assessment of urine flow rate can be difficult in ambulatory or free-living settings. Therefore, GFR is often further estimated based upon serum creatinine and equations incorporating corrections for age, sex, race and body size, thereby eliminating the need for urine collection [76]. As a result, a spot assessment of serum creatinine is often part of routine medical practice for assessment of kidney function [77]. It is also important to note that there is growing support for a spot assessment of circulating cystatin C as a marker of kidney function [78]. Cystatin C is produced at a stable rate by all cells within the body and freely filtered by the glomeruli. The potential benefit of using cystatin C is that changes in cystatin C during dynamic fluctuations in kidney function may occur much earlier than changes in serum creatinine [79]. In theory, this could provide for more rapid diagnoses of AKI [78]. Importantly, to our knowledge, cystatin C has never been used to quantify kidney function during exercise in the heat. Moreover, despite evidence that cystatin C may provide useful information, the clinical practice guidelines for the diagnosis and classification of the severity of AKI are currently based on acute changes in serum creatinine and/or absolute urine flow rate (Table 1).

The underlying basis for the AKI guidelines is the relation between serum creatinine and kidney function, and that rapid decreases in kidney function define AKI. A limitation regarding the use of changes in kidney function as diagnostic criteria for AKI (whether quantified via changes in serum creatinine, urine flow rate, or cystatin C), is that GFR is often acutely reduced as a result of an integrated physiological response (i.e., conditions extrinsic to the kidneys). This is often referred to as prerenal AKI [80], whereby increases in serum creatinine (and/or reductions in urine flow rate) may satisfy the definition of AKI, but these reductions in kidney function are due to neural, hormonal, and/or hemodynamic responses upstream of the kidneys. For example, GFR and urine flow rate are decreased during dehydration [81,82]. This is likely due to reductions in renal blood flow [83,84] caused by increases in renal sympathetic nerve activity [85,86], vasopressin release [87], and activation of the renin–angiotensin–aldosterone system [88]. These are normal and healthy physiological responses that promote fluid conservation [89]. Thus, increases in serum creatinine, which are normally reflective of a reduction in GFR, may not always be indicative of kidney injury during dehydration [80]. Because of this, it is often recommended that the creatinine and/or urine flow rate-based diagnostic criteria for AKI be used only after an optimal state of hydration has been achieved [73]. However, an optimal hydration state is ill-defined given that no single variable truly captures body fluid status [90].

Given the limitations of kidney function-derived AKI diagnoses, a rapidly growing body of literature aiming to identify biomarkers of AKI has emerged [91]. The goal of these biomarkers is to identify those individuals at risk of developing AKI before any reductions in kidney function occur [92,93,94,95]. It has recently been reported that over the past 10 years there have been 3300 scientific publications and hundreds of AKI biomarkers investigated [91]. The validity and clinical utility of many of these biomarkers remains to be determined, even for well-studied AKI biomarkers [91,96]. However, when combined with standard indices of kidney function, measurement of AKI biomarkers may provide unique information regarding interactions between changes in kidney function and the potential for AKI [96,97]. In the following, we will introduce a few promising AKI biomarkers, with an emphasis on those that have been used in experimental human subjects research, particularly as it relates to exercise, dehydration, and/or heat strain, or that demonstrate potential for use in laboratory-based (i.e., not clinical settings) human subjects studies. An in-depth overview of AKI biomarkers is outside of the scope of this review. Thus, to the interested reader we recommend a number of more comprehensive reviews addressing the pathophysiological bases and clinical performance of AKI biomarkers [92,93,94,95].

### AKI Biomarkers

*Neutrophil gelatinase-associated lipocalin* (NGAL) is expressed in multiple cell types (e.g., renal, hepatic, cardiac, etc.) at relatively low, but constant levels [98]. NGAL generally functions as a bacteriostatic agent [99]. Urinary NGAL is the most widely studied biomarker of AKI [94]. Renal NGAL mRNA and protein are strongly upregulated after ischemic or toxic kidney injury in both human and animal models [100,101,102,103]. In the kidneys, NGAL is produced in the thick ascending limb of the loop of Henle and intercalated cells of the collecting duct [104]. In addition, circulating NGAL from extrarenal sources is filtered by the glomeruli and reabsorbed in the proximal tubules [105]. The increased urinary NGAL concentrations in the context of AKI are likely caused by endogenous NGAL production in the kidneys and reductions in tubular NGAL reabsorption [94,104], a hypothesis supported by clinical evidence that urinary NGAL is not elevated in isolated prerenal AKI [33,106], although these findings are not unanimous [107]. Therefore, urinary NGAL appears to be a marker of general tubular injury, without a specific etiology [93,94,104] (Figure 1). Plasma NGAL may also have some utility as a biomarker of AKI [93]. However, increases in plasma NGAL likely more readily reflect reductions in GFR, as well as renal ischemia and/or glomerular dysfunction [93,95] (Figure 1).

*Kidney injury molecule-1* (KIM-1) is a transmembrane glycoprotein that is expressed at low levels in the normal kidney, but is further upregulated following ischemia-reperfusion and toxic kidney injury [108,109]. KIM-1 is mainly upregulated in proximal tubule cells during AKI [110,111]. Thus, increases in urinary KIM-1 likely indicate proximal tubule injury, although the etiology is nonspecific (Figure 1). Clinical studies investigating the utility of KIM-1 have shown variable results [94].

*Liver-type fatty acid binding protein* (L-FABP) is a cytoplasmic protein that protects against oxidative stress induced by peroxisomal metabolism [112], particularly in the presence of hypoxia given that the human L-FABP gene contains a hypoxia-inducible factor 1⍺ response element [113]. L-FABP is excreted by proximal tubule epithelia into the tubular lumen together with bound peroxisomal products [113]. Urinary L-FABP can likely be used to identify patients at risk of developing AKI. This is supported by data indicating that patients with high L-FABP levels measured at the time of intensive care unit admission had a greater risk for developing AKI within 1 week, compared to a group of patients with lower L-FABP levels [114]. The genetic link with hypoxia-inducible factor 1⍺, together with findings indicating that urinary L-FABP levels are strongly correlated with the duration of ischemia in kidney transplant recipients [113], suggests that urinary L-FABP may provide an indication that the mechanism of this risk of AKI is related to proximal tubule hypoxemia and the development of oxidative stress (Figure 1).

*Interleukin-18* (IL-18) is a proinflammatory cytokine that is produced in the intercalated cells of the collecting ducts of healthy kidneys [115], but is more broadly made in tubular epithelial cells as part of the inflammatory cascade induced by AKI [116]. Urinary IL-18 has been shown to be a promising biomarker of AKI in animal models [94]. Clinical studies also demonstrate that urinary IL-18 may have utility in predicting AKI [117]. However, its diagnostic value remains less clear [117]. Nevertheless, urinary IL-18 is likely a marker of general tubular injury and subsequent inflammatory-pathway activation (Figure 1).

*Insulin-like growth factor binding protein 7* (IGFBP7) and *tissue inhibitor metalloproteinase 2* (TIMP-2) are proteins known to induce G1 cell cycle arrest [118]. In general, cell cycle arrest likely occurs to prevent potential DNA damage during cellular stress [119]. However, if cells stay in a phase too long or exit a phase too soon, the normal division and repair process can become maladaptive [120]. Cell cycle arrest occurs in renal epithelial cells in a variety of in vitro models of AKI [121] and is associated with the development of fibrosis following multiple types of AKI in rodents [122]. The arithmetic product of urinary IGFBP7 and urinary TIMP-2 ([IGFBP7•TIMP-2]) was identified as an AKI biomarker in a clinical study in which it outperformed ~338 other candidate biomarkers at predicting AKI based on standard clinical criteria (Table 1) [123]. In 2014, an [IGFBP7•TIMP-2] test system (better known by its proprietary name, NEPHROCHECK^®^) received FDA approval for marketing as a screening tool to estimate the risk of AKI development [124]. FDA approval carefully stipulated that increases in [IGFBP7•TIMP-2] should not necessarily be interpreted to indicate that AKI is ongoing. This indication is consistent with the fact that G1 cell cycle arrest occurs during the very early stages of cellular stress [125]. Notably, [IGFBP7•TIMP-2] is the only biomarker approved by the FDA for an AKI related indication [124]. Recent evidence indicates that increases in urinary [IGFBP7•TIMP-2] following diverse forms AKI is contributed to by a multitude of factors, which include decreases in glomerular permeability, proximal tubular cell leakage, and impaired reabsorption of IGFBP7 and TIMP-2 in the proximal tubule [126]. Thus, increases in [IGFBP7•TIMP-2] can be interpreted as evidence of the potential for renal injury of a nonspecific origin occurring in the glomeruli and proximal tubules (Figure 1).

## 3. Interpretation of AKI Biomarkers in Non-Clinical Settings

Recent advances in our understanding of AKI biomarkers have unexpectedly resulted in reports of the potential for AKI in situations traditionally considered to be clinically benign for the kidneys. This is highlighted by studies consistently demonstrating increases in AKI biomarkers following a single bout of prolonged endurance exercise [127,128,129,130,131,132]. Such findings have raised the question as to whether regularly engaging in endurance exercise may lead to poor renal health outcomes [133]. Arguments against this position are numerous and highlighted by the supposition that since the development of the ‘jogging phenomenon’ in the early 1960s [134] there has not been a profound increase in the incidence of AKI. Rather, the epidemic of AKI in the general population is largely attributed to patients hospitalized with acute illnesses and those undergoing major surgery, which may be partially contributed to by a greater recognition of AKI, and improved diagnostic and classification criteria [135]. Moreover, regularly engaging in vigorous physical activity (such as endurance exercise) reduces the risk of developing CKD [136], suggesting that the repeated exposures to exercise-induced elevations in AKI biomarkers does not lead to long-term sequelae. This has raised the question regarding how to interpret acute increases in AKI biomarkers in otherwise healthy individuals. This is an important consideration with regards to understanding the risk of AKI occurring subsequent to exercise in the heat.

The AKI biomarkers described herein were developed to identify AKI as would occur in clinical situations, in which large and sustained elevations were expected. In such instances, elevations in these AKI biomarkers would be indicative of intrinsic renal damage occurring at various locations along the nephron (Figure 1) [96,137]. Data obtained from otherwise healthy individuals engaging in endurance exercise demonstrate consistent increases in AKI biomarkers [127,128,129,130,131,132], but these elevations are often shorter in duration and not to the same extent as those observed in clinical situations [100,101,102,103,113,117,123]. These consistent elevations in AKI biomarkers are often interpreted as meaningful, but because the increases are small, they are usually not interpreted as being indicative of intrinsic renal injury. Rather, these small increases in AKI biomarkers likely reflect an increased potential to develop of AKI, although the possibility that increases in AKI biomarkers reflect some degree of intrinsic injury cannot be completely ruled out at this time. We believe this conclusion is indirectly supported by evidence that elevations in AKI biomarkers in the laboratory setting typically resolve ~24 h following exercise [127,130]. Thus, small, transient increases in AKI biomarkers are likely indicative of acute kidney stress [138]. This state likely represents a period in which there is an increased risk of developing AKI, with the magnitude of this risk occurring in proportion to the magnitude of elevations in AKI biomarkers. Notably, this definition is consistent with the FDA approved indication for urinary [IGFBP7•TIMP-2] as a biomarker of the magnitude of the risk associated with developing AKI.

Currently, there is no consensus regarding the best AKI biomarker to measure. Urinary NGAL may have the most robust evidence base [94], but only urinary [IGFBP7•TIMP-2] has an FDA approved indication for AKI [124]. That said, to our knowledge no study has measured urinary [IGFBP7•TIMP-2] under conditions of exercise, heat strain and/or exercise. In this context, there is likely value in employing a battery of assays for AKI biomarkers. This approach may provide important insights regarding: (i) the magnitude of the risk of AKI, (ii) the anatomical location where this risk originated, and (iii) the potential etiology of this risk (Figure 1). Such an understanding may provide important information towards determining countermeasures to reduce the risk of AKI in the context of exercise in the heat.

There is likely also value in combining the measurement of AKI biomarkers with traditional indices of kidney function (e.g., serum creatinine, urine flow rate, etc.) for interpreting the risk of AKI. This approach has been proposed in the clinical literature as a method to discern prerenal AKI (i.e., only a reduction in kidney function without increases in AKI biomarkers) from intrinsic AKI (i.e., a reduction in kidney function with increased AKI biomarkers) or subclinical AKI (i.e., increased biomarkers in the absence of a reduction in kidney function) [96,97,137,139]. In a similar manner, the risk of AKI can likely be assessed in the context of exercise in the heat (Figure 2). For instance, a decrease in kidney function without increases in AKI biomarkers may be indicative of a relatively mild risk for AKI because the change in function is unlikely to be pathological. This decline in kidney function may be the normal physiological response of the kidneys to a stressor (such as exercise) that is completely reversed once the stress has abated. That said, increases in AKI biomarkers without a change in kidney function may be interpreted as a slightly higher (i.e., moderate) risk of AKI, owing to the presence of potential pathological processes. Finally, the highest risk of AKI may occur when kidney function is reduced alongside increases in AKI biomarkers. Moreover, this risk could be further delineated based on the magnitude of the reductions in kidney function and/or increases in AKI biomarkers. For example, the condition invoking the greatest reductions in kidney function and largest increases in AKI biomarkers would be interpreted as having the highest risk of AKI. Notably, however, this AKI risk matrix is likely only useful in situations where comparisons between conditions might be considered the most important (i.e., it may only provide an index of relative AKI risk). The utility of this matrix for providing information regarding absolute AKI risk is currently unclear.

Finally, an important methodological consideration is the normalization of urinary AKI biomarkers to urine concentration. It is common in clinical and research settings to correct urine-based AKI biomarkers by normalizing to urinary creatinine [140]. However, the application of urinary creatinine normalization in exercise models is likely flawed [36]. This is because urinary creatinine excretion becomes inconsistent both within and between subjects during and following exercise, owing to relatively large swings in GFR [141]. Thus, in controlled laboratory settings it is often suggested that the most accurate method to quantify renal biomarkers requires the collection of timed urine specimens to estimate the excretion rate of a given biomarker [140]. Therefore, concentrations of urinary AKI biomarkers in the context of exercise in the heat are often normalized to urine flow rate [30,36]. That said, normalization to other markers of urinary concentration (e.g., osmolality) may also be a valid approach [131,132].

## 4. AKI Susceptibility Evoked by Exercise in the Heat in Humans

Clinical AKI associated with exercise in the heat has occasionally been reported in the clinical literature. To our knowledge, the most complete clinical dataset was published in 1967, which reported the cases of eight previously healthy military recruits who had developed AKI during training exercises outdoors in the summer months [142]. The overall incidence of AKI is generally very low in this population of military recruits. However, it was later estimated that ~10% of the AKI cases treated at Walter Reed General (Military) Hospital from 1960 to 1966 were due to AKI occurring subsequent to exercise in the heat [143]. Thus, there is clinical evidence that exercise in the heat can bring about AKI, at least in a subset of individuals.

More recently, traditional measures of kidney function have been combined with AKI biomarkers to examine the risk of AKI during exercise in the heat. Generally, the findings from these studies suggest that exercise in the heat can increase the susceptibility to AKI. For instance, Junglee et al. demonstrated that mild heat strain (+1.3 °C increase in core temperature) and mild dehydration (~1% body weight loss) due to exercise in the heat resulted in elevations in serum creatinine, reductions in urine flow rate, and increases in plasma NGAL [36]. Our laboratory has furthered this work and found that elevations in serum creatinine and plasma NGAL were influenced by the magnitude of increases in core temperature and the extent of dehydration produced by exercise in the heat of two different durations [29] (Figure 3). Of the increases in serum creatinine, ~30% of the observations satisfied the criteria for stage 1 AKI according to the AKIN criteria (Table 1) [73] (Figure 3). We also showed that elevations in serum creatinine and plasma NGAL returned to baseline levels the following day [29] (Figure 3), which is supportive of the idea that this period of an increased risk of AKI is transient. Similarly, McDermott et al. found that 4–6 h of exercise in the heat, which evoked moderate dehydration (~1.6% body weight loss), elevated both serum creatinine and serum NGAL [70]. Collectively, the findings to date support that exercise in the heat can increase the risk of AKI in humans and that this increased risk occurs in proportion to the magnitude of heat strain and dehydration. To our knowledge, however, no study in the context of exercise in the heat has systematically examined markers of kidney function simultaneous with a panel of AKI biomarkers. This is a significant limitation with regards to the strength of the evidence base. Thus, future studies are necessary to conclusively determine the risk of AKI caused by exercise in the heat.

Importantly, many of the potential modulators of the magnitude of AKI risk evoked by exercise in the heat remain largely unexplored. This is important because resolving these unknowns will have important ramifications regarding the development of countermeasures to alleviate the risk of AKI during exercise in the heat. For instance, the relative importance of heat strain versus dehydration on the magnitude of the risk of AKI evoked by exercise in the heat is unknown. Although a formal body of evidence is lacking, there is evidence that hydration status may be a comparatively more important modulator of AKI risk compared to heat strain alone. For instance, allowing rodents full access to water during 4 weeks of repeated heat exposure prevented the renal injury and nephropathy that occurred in the rodents that were exposed to the same heat load, but were restricted from drinking during the heat exposures [47]. Thus, it can be speculated that hydration status is the most important modulator of AKI risk during exercise in the heat. That said, a limitation of this conclusion is that core temperature is often elevated to a greater extent in a dehydrated state during exercise and/or heat exposure [144]. In the aforementioned study, core temperature in the rats was not measured [47]. Therefore, the comparative importance of heat strain versus dehydration on the risk of AKI during exercise in the heat remains largely uncertain. Another important knowledge gap is the effect of substances that have nephrotoxic side effects on the risk of AKI during exercise in the heat. The best example is likely NSAIDs, which are commonly used in occupational settings [18,19]. Notably, a 1.2 g dose of the NSAID ibuprofen results in greater reductions in GFR during exercise in the heat compared to a placebo condition [145]. However, recent evidence indicates that this deleterious effect on kidney function does not translate to a greater risk of AKI, such that a 0.6 g dose of ibuprofen did not lead to greater increases in serum NGAL following 4–6 h of exercise in the heat [70]. However, these findings may be explained by the lower NSAID dose (0.6 vs. 1.2 g). Thus, future studies are required to elucidate the modulatory role of nephrotoxic substances (particularly NSAIDs) on the risk of AKI during exercise in the heat.

### Mechanisms of AKI Susceptibility Evoked by Exercise in the Heat

In general, the mechanisms by which exercise in the heat may increase the risk of AKI in humans remain largely unexplored. That said, data from animal models, together with some fundamental physiological data in humans, suggest that the potential increased risk of AKI provoked by exercise in the heat is multifactorial.

Heat strain [145,146], dehydration [81,82] and exercise [147,148,149] independently reduce renal blood flow in humans, and even more profound reductions in perfusion are observed when these conditions are combined [150]. These reductions in renal perfusion are largely due to increases in renal sympathetic nerve activity [151] and circulating vasopressin levels [152], but may also be contributed to by activation of the renin–angiotensin–aldosterone system. For instance, angiotensin II has vasoconstrictor actions in the kidneys [153] and may modulate kidney function during exercise in the heat in humans [154]. Notably, these declines in renal perfusion typically resolve with recovery [36,145,146,150] and are generally considered to be clinically inconsequential [16]. That said, data from dehydrated rats indicate that reductions in overall renal perfusion can invoke a heterogenous distribution of blood flow within the kidneys [84]. Strikingly, in this same study a sub-analysis of data from one rat using a different experimental technique indicated that dehydration can produce localized ischemia, particularly in cortical regions [84]. These data are seemingly corroborated by data in dogs whereby heat strain-induced reductions in renal perfusion were solely caused by reductions in cortical blood flow, without a change in medullary blood flow [155]. This relatively low blood flow state can compromise oxygen delivery to the renal cortex, which could provoke ischemia and a localized reduction in ATP [84]. Importantly, a low ATP environment can theoretically increase the risk of AKI secondary to increased oxidative stress and inflammation [156].

Against this background, it is clearly important to understand how exercise in the heat may affect renal vascular control in humans, particularly as it relates to intrarenal blood flow distribution and oxygenation. To our knowledge, such studies have never been carried out. This knowledge gap may be important for understanding the mechanisms by which exercise in the heat may increase the risk of AKI. For instance, the renal cortex, which demonstrates reductions in blood flow during dehydration (in rats) [84] and heat strain (in dogs) [155], anatomically houses the majority of the proximal tubules [157], where upwards to 65% of sodium is reabsorbed [158]. Sodium reabsorption in the kidneys can be directly stimulated by aldosterone [159], angiotensin II [160] and/or increases in renal sympathetic nerve activity [161], all of which are increased with heat strain, dehydration and/or exercise in an effort to maintain fluid homeostasis [162,163,164]. Sodium reabsorption is energetically expensive, owing to the reliance on the Na^+^/K^+^ pump [165]. Thus, an increased demand for sodium reabsorption may increase the susceptibility of the kidneys to injury particularly in the presence of additional stressors that can compromise ATP production (e.g., low blood flow). This is supported by data demonstrating that blocking the actions of aldosterone via the administration of a mineralocorticoid receptor antagonist attenuates the severity of injury in a rat model of mild ischemia-induced AKI [166], and prevents the subsequent development of CKD in a similar rodent model of ischemic AKI [167]. Aldosterone exerts its sodium reabsorption actions via the Na^+^/K^+^ pump [159]. Thus, these data support the idea that increased energy demands associated with activation of the Na^+^/K^+^ pump can contribute to the incidence and severity of AKI. Whether this holds true in the context of exercise in the heat remains unknown. Notably, sodium reabsorption is greater during exercise in the heat compared to passive heat exposure to the same extent of dehydration and heat strain [164], suggesting that exercise in the heat evokes a relatively large renal ATP demand. Moreover, sodium reabsorption is greater during exercise in the heat in the presence of dehydration [168]. This increased ATP demand is further challenged by the relatively low renal cortical oxygen delivery that is likely occurring subsequent to heat strain, dehydration and exercise. Oxygen delivery is required to sustain ATP production. Moreover, any mismatch between oxygen delivery and oxygen demand increases the susceptibility of the kidneys to injury [169], which likely occurs subsequent to increased oxidative stress and inflammation [156]. Thus, it is possible that a redistribution of blood flow within the kidneys, together with an increased oxygen demand, contributes to the increased risk of AKI during exercise in the heat. However, direct evidence is warranted.

There is certainly reason to believe that reductions in renal blood flow contribute to the increased risk of AKI during exercise in the heat in humans. However, there are two arguments against this contention. First, heat acclimation (or acclimatization) invoked by repeated heat exposures stimulates greater reductions in renal blood flow during heat exposure in rats [170], which contributes to improvements in heat tolerance [171]. Recent evidence, however, indicates that heat acclimation over a 23 day period attenuated the rise in serum creatinine during exercise in the heat in humans [172]. This was interpreted as evidence that heat acclimation likely has a protective effect in the kidneys, such that the incidence of serum creatinine defined AKI was attenuated with heat acclimation [172]. Despite the limitations associated with using serum creatinine to define AKI risk, such findings suggest that the presumed greater reductions in renal blood flow during exercise in the heat following heat acclimation may not translate to a greater risk of AKI. That said, it is important to note that workers experiencing CKDu are likely heat acclimated. Thus, the presumed protective effect of heat acclimation on kidney health remains uncertain. Second, older adults have attenuated reductions in renal blood flow during exercise in the heat [173] and passive heat exposure [174]. However, the incidence of AKI during heat waves is disproportionately higher in older adults compared to younger adults [175,176,177,178]. Thus, the theoretical benefit conferred by the relative maintenance of renal blood flow during heat exposure may not be protective against the risk of AKI in older adults. That said, these findings may be confounded by other age-related changes in kidney function, such as overactivation of the polyol-fructokinase pathway [179] and augmented vasopressin responses [180,181], both of which are discussed below. Nonetheless, there is a clear need to discern the role of reductions in renal blood flow to the increased risk of AKI during exercise in the heat in humans.

Rodent models also convincingly demonstrate that nephropathy due to recurrent AKI elicited by repeated heat exposures is contributed to by activation of the intrarenal polyol-fructokinase pathway and the endogenous production of fructose [47,48,49,182,183]. The polyol pathway is stimulated by an increased plasma osmolality that upregulates the enzyme aldose reductase [184]. Aldose reductase catalyzes the conversion of glucose into sorbitol, which has a protective effect against a hyperosmolar renal environment [184]. However, this benefit is not sustainable. Recent evidence has uncovered the potential deleterious effect of the subsequent activation of the fructokinase pathway, particularly with recurrent heat exposure and/or dehydration [185]. Under such conditions, sorbitol can be metabolized to fructose by sorbitol dehydrogenase and this fructose is then broken down by the enzyme fructokinase [185]. The metabolic breakdown of fructose occurs rapidly and is energetically costly [186]. Thus, activation of the fructokinase pathway can further reduce ATP availability [187]. It should be noted that the anatomical location of the polyol-fructokinase pathway is the proximal tubules [185,186], a location that likely has experienced a selective reduction in blood flow [84,155] and probably already has a heightened ATP demand [165] due to sodium reabsorption [164,168] and the activation of the Na^+^/K^+^ pump [159]. Collectively, this reduced ability to generate ATP can promote oxidative stress and inflammation [156] and can stimulate uric acid production [186]. Notably, hyperuricemia can further reduce renal perfusion [188] and independently can incite the polyol-fructokinase pathway [49,189], thereby exacerbating the oxidative stress and inflammation [67] that ultimately causes AKI [156]. Importantly, the fructokinase pathway is at least partially mediated by actions associated with vasopressin release [48,183,190], the production of which is stimulated by increases in plasma osmolality and/or hypovolemia [191].

The importance of the polyol-fructokinase pathway in AKI associated with heat exposure is highlighted by data indicating that when rats consume a high fructose beverage, which exogenously increases substrate for the fructokinase pathway, the resulting AKI and kidney damage from recurrent heat exposure is exacerbated [182,183]. Moreover, when the ability to generate fructokinase is genetically knocked out, mice exposed to recurrent heat exposure do not demonstrate AKI or kidney damage [47]. We have recently provided similar evidence in humans such that drinking a soft drink with a high fructose content during and following 4 h of exercise in the heat exacerbates increases in serum creatinine, reductions in urine flow rate, and elevations urinary NGAL, compared to when drinking an equivalent amount of water [30] (Figure 4). Interestingly, this observation occurred alongside greater increases in copeptin (a stable surrogate for vasopressin) and serum uric acid (Figure 4). These latter findings support the animal data [48,182,183] and the mechanisms by which the activation of the polyol-fructokinase pathway may increase the risk of AKI [185]. This recent work suggests that polyol-fructokinase pathway activity may modulate kidney function and the risk of AKI following a single bout of exercise in the heat in humans. It is important to note, however, that to our knowledge there is no evidence regarding whether the polyol-fructokinase pathway directly modulates the risk of AKI during exercise in humans. This is likely due to a lack of established biomarkers for determining the activation of these pathways in humans. Further work is required to establish such biomarkers and to apply this knowledge to exercise in the heat.

## 5. Summary

The purpose of this narrative review was to present evidence that exercise in the heat increases the risk of developing AKI in humans. A growing body of evidence demonstrates that passive heat exposure in rodents causes kidney injury, and when this injury is chronically repeated it is capable of eliciting CKD. Experimental evidence in humans is more limited, with many gaps in the literature (Table 2). That said, studies consistently demonstrate that a single bout of exercise in the heat increases biomarkers of AKI and that the magnitude of these elevations in AKI biomarkers is dependent on the extent of heat strain and dehydration. In this context, elevations in AKI biomarkers are not necessarily indicative of intrinsic renal damage. Rather, a better interpretation is that they reflect an increased risk of AKI. Data largely derived from animal models indicate that the mechanism(s) underlying elevations in AKI biomarkers is multifactorial (Figure 5). For instance, heat-related reductions in renal blood flow may provoke a heterogenous blood flow distribution within the kidneys. In theory, this can promote localized ischemia, hypoxemia and ATP depletion, which may be exacerbated by an increased demand for sodium reabsorption—an energetically expensive process. Moreover, heightened polyol-fructokinase pathway activity likely exacerbates tissue hypoxemia and ATP depletion, occurring secondary to the intrarenal production of fructose and hyperuricemia. Collectively, these responses likely promote inflammation and oxidative stress, which can elevate biomarkers of AKI in otherwise healthy humans. Unfortunately, there is currently very little direct mechanistic evidence to support how exercise in the heat may increase the risk of AKI in humans (Table 2). This is an important knowledge gap that has ramifications regarding the development of countermeasures to safeguard the renal health of people who regularly engage in physical work in hot environments.

## Figures and Tables

**Figure 1 nutrients-11-02087-f001:**
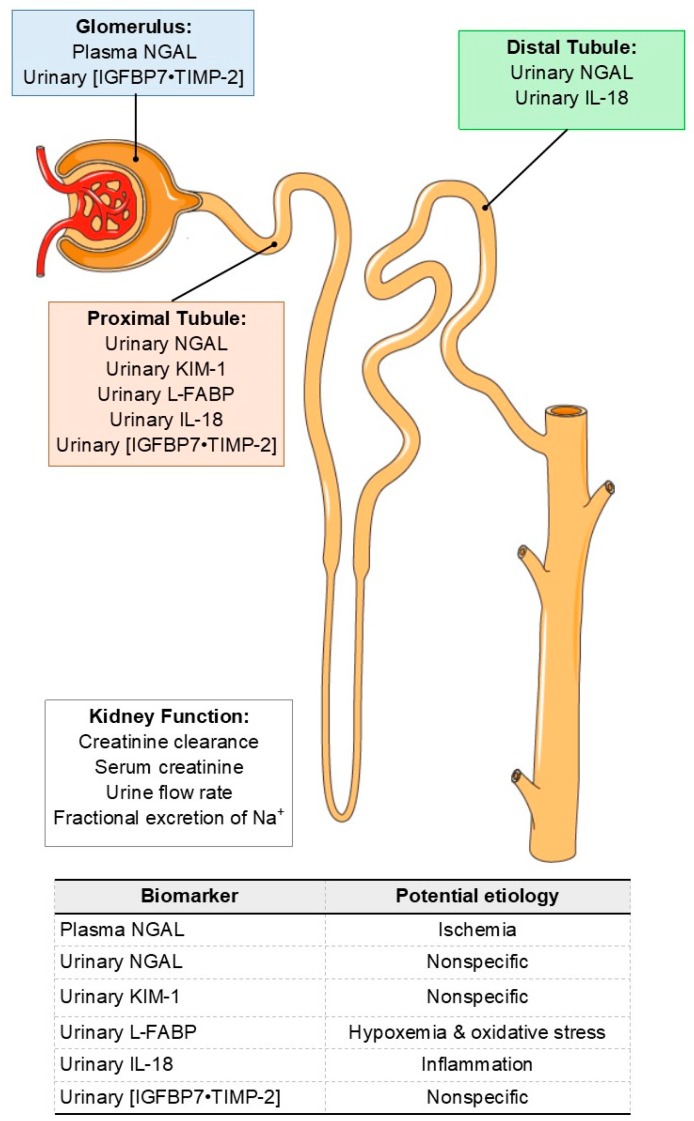
Top: Anatomical locations for biomarkers indicative of an increased risk of acute kidney injury (AKI) and common clinically relevant measures indicative of overall kidney function. Bottom: Potential etiology underlying increases in AKI biomarkers. Abbreviations—NGAL: Neutrophil gelatinase-associated lipocalin, [IGFBP7•TIMP-2]: the product of Insulin-like growth factor binding protein 7 (IGFBP7) and tissue inhibitor metalloproteinase 2 (TIMP-2), KIM-1: Kidney injury molecule-1, L-FABP: Liver-type fatty acid binding protein, IL-18: Interleukin-18. Please refer to text for references.

**Figure 2 nutrients-11-02087-f002:**
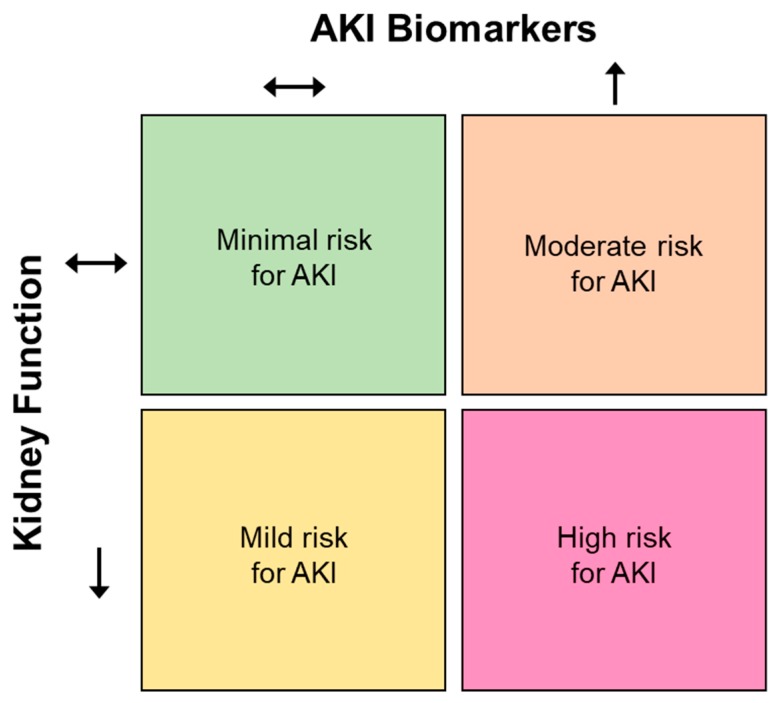
Proposed method for determining the relative risk of developing acute kidney injury (AKI) during exercise in the heat based on changes in kidney function and AKI biomarkers. A decrease in kidney function without increases in AKI biomarkers may be indicative of a relatively mild risk for AKI because the renal changes are unlikely to be pathological (bottom-left). Increases in AKI biomarkers without a change in kidney function may be interpreted as a slightly higher (i.e., moderate) risk of AKI, owing to the presence of potential pathological processes (top-right). The highest risk of AKI may occur when kidney function is reduced alongside increases in AKI biomarkers (bottom-right). AKI risk could be further delineated based on the magnitude of the reductions in kidney function and/or increases in AKI biomarkers.

**Figure 3 nutrients-11-02087-f003:**
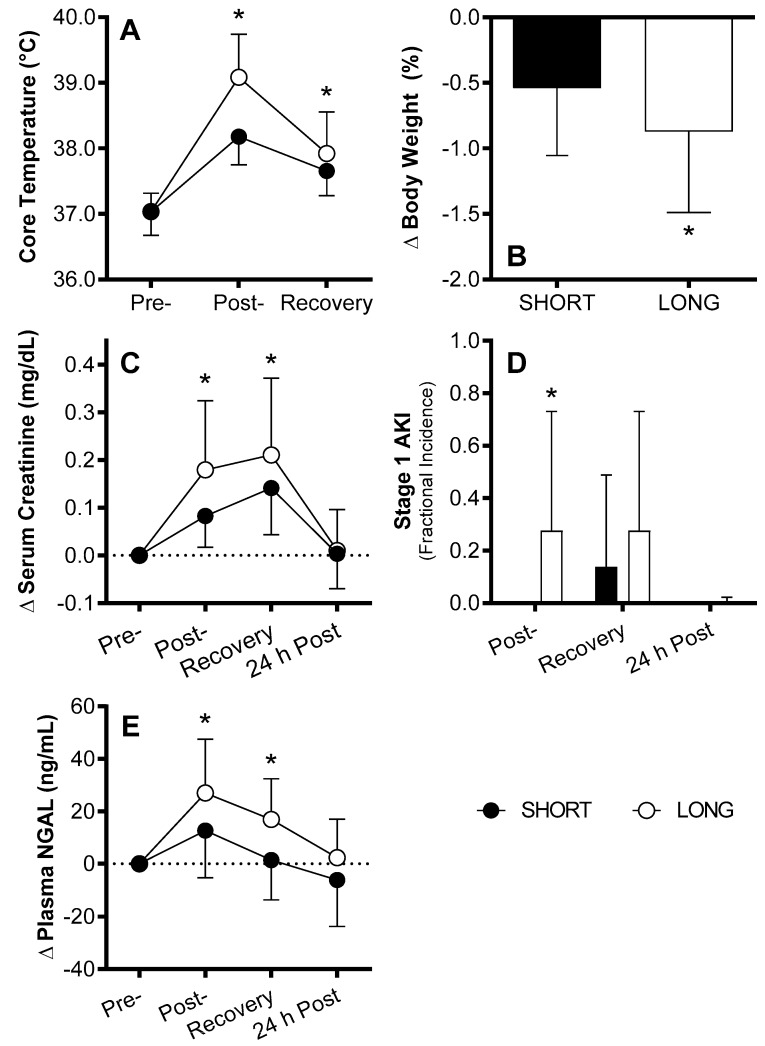
During the longer of two bouts of exercise in the heat (LONG vs. SHORT), greater heat strain (**A**) and dehydration (**B**) resulted in greater changes (Δ) in serum creatinine (**C**), which were sufficient to satisfy the criteria for Stage 1 acute kidney injury (AKI) in ~30% of the subjects (**D**), and greater increases in plasma neutrophil gelatinase-associated lipocalin (NGAL) (**E**). * different from SHORT (*p* < 0.05), Mean ± SD, n = 29. From Schlader et al. [29] with permission.

**Figure 4 nutrients-11-02087-f004:**
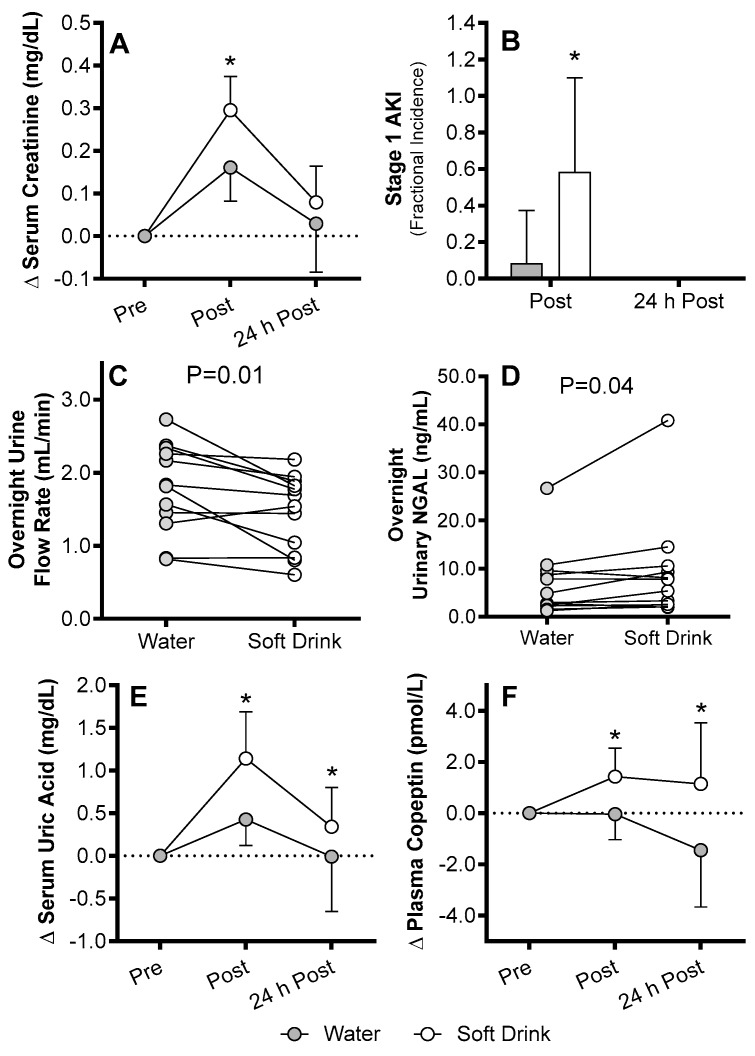
Despite no differences in core temperature or changes in body weight (data not shown), drinking a high fructose soft drink compared to water during and following 4 h of exercise in the heat resulted in greater changes (Δ) in serum creatinine (**A**), meeting the criteria for Stage 1 acute kidney injury (AKI) at Post exercise in ~60% of the subjects (**B**). During the overnight period (defined as the time from leaving the laboratory immediately after post-exercise data collection until returning to the laboratory 24 h following pre-exercise (~18 h following post-exercise)), drinking a soft drink reduced urine flow rate (despite drinking ~347 mL more fluid during the overnight period) (**C**) and elevated urinary neutrophil gelatinase-associated lipocalin (NGAL) (**D**). These changes in indices in kidney function and biomarkers of AKI in the soft drink trial were paralleled by greater elevations in serum uric acid (**E**) and plasma copeptin (**F)**, a stable surrogate for vasopressin. * different from Water (*p* < 0.05), Mean ± SD or individual values, n = 12. From Chapman et al. [30] with permission.

**Figure 5 nutrients-11-02087-f005:**
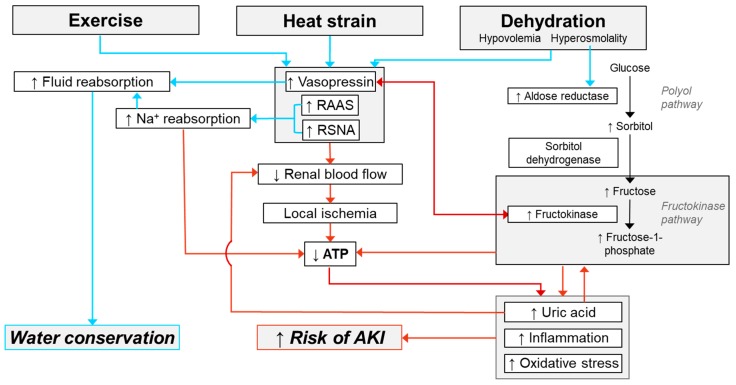
Potential mechanisms by which exercise in the heat and the subsequent development of heat strain (i.e., increases in core body temperature) and dehydration (a hypovolemic, hyperosmotic state) may increase the risk of acute kidney injury (AKI) while simultaneously promoting fluid conservation. Red arrows indicate potential pathophysiological pathways. Blue arrows indicate known beneficial physiological responses. Abbreviations—RAAS: Renin–angiotensin–aldosterone system, RSNA: Renal sympathetic nerve activity, ATP: adenosine triphosphate. Please refer to text for references.

**Table 1 nutrients-11-02087-t001:** Clinical criteria for identifying acute kidney injury and staging the severity of injury.

	Criteria	
	Serum Creatinine	Urine Output
**Acute Kidney Injury Network (AKIN) Classification**		
Stage 1	Increase ≥0.3 mg/dL (≥26.5 µmol/L) OR increase ≥1.5–2.0-fold from baseline	<0.5 mL/kg/h for 6 h
Stage 2	Increase >2.0–3.0-fold from baseline	<0.5 mL/kg/h for 12 h
Stage 3	Increase >3.0-fold from baseline OR serum creatinine ≥4.0 mg/dL (≥354 µmol/L) with an acute increase of ≥0.5 mg/dL (44 µmol/L) OR need for renal replacement therapy	<0.3 mL/kg/h for 24 h OR anuria for 12 h OR need for renal replacement therapy
**Kidney Disease: Improving Global Outcomes (KDIGO) Classification**		
Stage 1	Increase ≥0.3 mg/dL (≥26.5 µmol/L) OR 1.5–1.9 times baseline	<0.5 mL/kg/h for 6-12 h
Stage 2	2.0–2.9 times baseline	<0.5 mL/kg/h for 12 h
Stage 3	3.0 times baseline OR increase in serum creatinine to ≥4.0 mg/dL (≥354 µmol/L) OR need for renal replacement therapy OR in patients <18 years old a decrease in eGFR to <35 mL/min/1.73 m^2^	<0.3 mL/kg/h for 24 h OR anuria for 12 h
**Acute Dialysis Quality Initiative (ADQI):** **Risk, Injury, Failure, Loss of Kidney Function and End-Stage Kidney Disease (RIFLE) Classification**		
Risk	1.5-fold increase OR GFR decrease >25% from baseline	<0.5 mL/kg/h for 6-12 h
Injury	2.0-fold increase OR GFR decrease >50% from baseline	<0.5 mL/kg/h for 12 h
Failure	3.0-fold increase OR GFR decrease >75% from baseline OR serum creatinine ≥4.0 mg/dL (≥354 µmol/L) with an acute increase of ≥0.5 mg/dL (44 µmol/L)	<0.3 mL/gk/h for 24 h OR anuria for 12 h
Loss of kidney function	Complete loss of kidney function >4 weeks	
End-stage kidney disease	Complete loss of kidney function >3 months	

Abbreviations: GFR: glomerular filtration rate, eGFR: estimated glomerular filtration rate. Please refer to text for references.

**Table 2 nutrients-11-02087-t002:** Selected identified knowledge gaps regarding the link between exercise in the heat, acute kidney injury and the development of chronic kidney disease in humans.

• **Acute Kidney Injury and the Development of Chronic kidney Disease:**
• Can exercise in the heat induce intrinsic renal damage?
• How do we interpret increases in AKI biomarkers associated with exercise in the heat?
• Can repeated exposures to AKI caused by exercise in the heat lead to CKDu? How does the frequency and severity of this AKI relate to the development and severity of CKDu?
• Does heat acclimation alleviate the risk of AKI (and CKDu) associated with exercise in the heat?
**Mechanisms by Which Exercise in the Heat May Increase the Risk Of Acute Kidney Injury:**
• What is the relative importance of heat strain versus dehydration on the magnitude of the risk of AKI evoked by exercise in the heat?
• Do NSAIDs (or other common substances with nephrotoxic side effects) modulate the risk of AKI evoked by exercise in the heat?
• To what extent does exercise in the heat invoke a heterogenous distribution of blood flow in the kidneys? What are the contributions of heat strain and/or dehydration?
• Do reductions in renal blood flow during exercise in the heat cause localized ischemia, reductions in oxygenation, and/or decreases in ATP availability? Where do these changes occur within the kidneys?
• Does exercise in the heat promote inflammation and oxidative stress within the kidneys? What are the contributions of heat strain and/or dehydration?
• What is the extent by which activation of the Na^+^/K^+^ pump contributes to the risk of AKI during exercise in the heat?
• Does the polyol-fructokinase pathway directly contribute to the risk of AKI during exercise in the heat?
• What are the roles of vasopressin and hyperuricemia in the risk of AKI during exercise in the heat?

Abbreviations: AKI: acute kidney injury, CKDu: chronic kidney disease of unknown origins, NSAIDs: nonsteroidal anti-inflammatory drugs.

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
