# Peer review of "The Potential for Renal Injury Elicited by Physical Work in the Heat"

_nutrients, 2019, doi:10.3390/nu11092087_

Round 1
Reviewer 1 Report
This is an excellent review by leaders in the field. I enjoyed reading the review and agree that the focus on renal ATP levels may be of key importance.
I only found one minor error in figure 4- -uric acid should be in mg/dl
I wonder if a table of next steps would be helpful. What needs to be done to determine the importance of these pathways-- to me it seems we need to see if there is a link between mild AKI and later development of CKD or hypertension. Is there any evidence from the military literature?
Reviewer 2 Report
Acording to Nutrients Instructions for authors, this narrative review does not provide concise and precise updates on the latest progress made in a given area of research. In addition, the text does not specify the methodology that has been followed to search and summarize the evidence analyzed in this narrative review. Furthermore, the authors should follow the PRISMA guidelines.
When the abstract is read, you realized that in it abstract does not just reflect the results that are budgeted as important in the title.
Finally, I wonder why there is a summary section at the end of the text. Are you highlighting the most important ideas and future lines of research or is an enhanced replica of the abstract?
Reviewer 3 Report
I read this comprehensive review with interest .
I had a dificult agenda time and asked a friend which whom I collaborate closely on CKD u we call CINAC ( chronic interstitial nephritis in agricultural communities) to help me in this review . He is the first author of our group of a paper in Kidn Int r who clearly summarized the available argiuments against the major role of heat stress .dehydration in the etiology of CINAC
Hereby his report
Hot, tired and thirsty: the potential for renal injury elicited by physical work in the heat- Review
Background
Up to line 125 it’s very rational and well argued.
“----epidemiological data obtained from Guatemalan sugarcane workers indicate that the presence of rhabdomyolysis is not associated with cross-shift declines in kidney function, suggesting that the contribution of rhabdomyolysis to progressive reductions in kidney function in this population is likely small (line 77-79). Therefore, the contribution of subclinical rhabdomyolysis to heat-related AKI and CKDu is likely minimal.” (Lines 84-85).
“It is important to note that there is currently no direct support for recurrent heat-related AKI in the etiology of CKDu. For instance, to our knowledge a dose-response relation between the frequency and severity of heat-related AKI and the subsequent development of CKD has never been experimentally examined in rodent models nor explored in epidemiological studies.” (Lines 86-89)
AKI (lines 126-191)
can be found in any textbook of physiology. I feel it’s redundant here and does not deserve a place in a specialized journal.
AKI biomarkers (Lines 192-258)- It’s a repetition of any review on biomarkers- all very well known. Authors could have made it a shorter description of biomarkers enough to make their point on diagnosis of AKI. Currently, it’s overlong. Much of the information given in this section is repeated in the next section.
Interpretation of AKI biomarkers in non-clinical settings
A very balanced and informative review. I thought it’s well argued. (Lines 259-344)
AKI susceptibility evoked by exercise in the heat in humans
Shows that Exercise in heat can cause AKI. It’s all well known. However, their comparison of the importance of heat strain and dehydration on AKI is interesting and thought-provoking.
Mechanisms of AKI susceptibility evoked by exercise in the heat
I thought this section is good. I can’t find anything wrong with it. But it only provides conjectures and extrapolations from animal studies which postulates the mechanisms of heat/dehydration induced AKI.
Both the abstract and the summary highlights the possible AKI to CKD transition caused by dehydration in CKDu. However, this paper does not show any evidence for a ‘heat induced AKI” to CKD continuum as postulated in the heat stress/dehydration theory of Johnson, et al. In the physiopathology sections of the discussion, the mechanism of AKI to CKD is not discussed at all. The “CKD” is mentioned only once in the main body of the discussion (line 458).
This paper gives a reasonably good review of AKI caused by heat but does not show any evidence for heat/dehydration being responsible for CKDu despite their claims in the abstract and the summary.
29/07/2019
I agree with what is written by him
I want to add that ref 23 is not quoted in a correct way
THe following recent ref should be quoted and commented
Chapman, E. et al. Risk factors for chronic kidney disease of non-traditional causes: a systematic review. Rev. Panam. Salud Pública (2019). doi:10.26633/rpsp.2019.35
Reviewer 4 Report
I congratulate the authors for this objective overview of the literature. I only had a few suggestions about adding additional information that are issues in epidemiological population-based studies on this topic.
Authors might want to mention the utility of cystatin c in determining the change in kidney function and AKI since acute changes in creatinine may not represent injury specifically in dehydrated study populations working/exercising in the heat. Also, mentioning whether or not there is a lag time with the creatinine and AKI biomarkers . Time of collection depends on whether you see an increase in biomarkers. For example, NGAL seems to spike earlier than creatinine (Figure 3). What effect does taking an NSAID (or other nephrotoxin) have on these biomarkers since a lot of these study populations take NSAIDs + heat + dehydration? Finally, the first part of the title, while catchy, doesn’t reflect the manuscript.
Round 2
Reviewer 2 Report
The manuscript has improved considerably and an adequate response has
been given to the comments of previous reviewers.
Author Response
Thank you.
Reviewer 3 Report
THis manusript remains confuse, sometimes simply incorrect ( wrong way to describe the content of particular key papers) biased, redundant on crtain pages,
The reader will be confronted an overall confusing impression. THe authors don't have the honesty to describe the different strong arguments why heat stress/dehydration is a marginal phenomenon if any in the etiology of CKDu.Furthermore the cite but in a wrong way an important review which clearly demonstrate that indeed ' there was no consistent evidence from our systematic review to support the association between CKDand heat stress-dehydration" and " We found consistent evidence for the adverse effect of agrochemicals on CKD and an association with end -stage renal failure "
